# Optogenetic stimulation of G$_s$-signaling in the heart with high spatio-temporal precision

Philipp Makowka[1], Tobias Bruegmann [1,2,4], Vanessa Dusend[1,2], Daniela Malan[1], Thomas Beiert[3], Michael Hesse[1], Bernd K. Fleischmann[1] & Philipp Sasse [1]

The standard technique for investigating adrenergic effects on heart function is perfusion with pharmaceutical agonists, which does not provide high temporal or spatial precision. Herein we demonstrate that the light sensitive G$_s$-protein coupled receptor JellyOp enables optogenetic stimulation of G$_s$-signaling in cardiomyocytes and the whole heart. Illumination of transgenic embryonic stem cell-derived cardiomyocytes or of the right atrium of mice expressing JellyOp elevates cAMP levels and instantaneously accelerates spontaneous beating rates similar to pharmacological β-adrenergic stimulation. Light application to the dorsal left atrium instead leads to supraventricular extrabeats, indicating adverse effects of localized G$_s$-signaling. In isolated ventricular cardiomyocytes from JellyOp mice, we find increased Ca$^{2+}$ currents, fractional cell shortening and relaxation rates after illumination enabling the analysis of differential G$_s$-signaling with high temporal precision. Thus, JellyOp expression allows localized and time-restricted G$_s$ stimulation and will provide mechanistic insights into different effects of site-specific, long-lasting and pulsatile G$_s$ activation.

[1] Institute of Physiology I, Medical Faculty, University of Bonn, 53127 Bonn, Germany. [2] Research Training Group 1873, University of Bonn, 53127 Bonn, Germany. [3] Department of Internal Medicine II, University Hospital Bonn, University of Bonn, 53127 Bonn, Germany. [4] Present address: Institute of Cardiovascular Physiology, University Medical Center, 37077 Göttingen, Germany. These authors contributed equally: Philipp Makowka, Tobias Bruegmann, Vanessa Dusend.  Correspondence and requests for materials should be addressed to P.S. (email: philipp.sasse@uni-bonn.de)

**N**euronal and humoral stimulation of β-adrenergic receptors activates $G_s$-proteins in cardiomyocytes in order to adapt heart function to increased oxygen demand of the body. The key signaling pathway is activation of adenylyl cyclases by the $G_\alpha$-subunit of the heterotrimeric $G_s$-protein leading to increased production of cyclic adenosine monophosphate (cAMP) which in turn stimulates the protein kinase A to phosphorylate target proteins[1–3]. This pathway leads to physiological elevation of the cardiac output by increasing beating rate and stroke volume. In contrast, chronic stimulation of the $G_s$-pathway can have detrimental effects by induction of cardiac hypertrophy and heart failure[2–5] as well as by enhancing the propensity for arrhythmia, especially in diseased hearts[2–4,6]. Current technologies to activate G-protein signaling require perfusion with receptor agonists, which inherently lack cell specificity and only provide limited temporal and spatial precision due to slow and uncontrollable diffusion and wash out kinetics[7]. In this report, we characterize in detail the use of the opsin from the lower invertebrate prebilaterian animal Jellyfish *Carybdea rastonii*[8], shortly termed "JellyOp", for optogenetic control of the $G_s$-signaling cascade in cardiomyocytes in vitro and in the intact heart. Importantly, JellyOp was previously shown to selectively activate $G_s$-proteins only, without promiscuous effects on $G_i$- or $G_q$-proteins[9].

## Results and discussion

**JellyOp activates $G_S$-signaling in cardiomyocytes.** JellyOp and GFP were expressed in cardiomyocytes under control of the ubiquitously active chicken β-actin promoter (Fig. 1a) by generation and differentiation of a stable transgenic G4 mouse embryonic stem cell (ESC) line. Transgenic JellyOp ESCs showed cytosolic GFP fluorescence and membrane-bound staining against the 1D4 epitope tag of JellyOp (Fig. 1b). Spontaneously beating embryoid bodies (EBs) generated from JellyOp ESCs contained α-actinin and GFP positive cardiomyocytes indicating JellyOp expression (Fig. 1c). Stimulation of JellyOp EBs with blue light (470 nm, 2.9 mW mm$^{-2}$, 5 min) led to an increase in cAMP levels to 646% of baseline, which was similar to maximal pharmacological stimulation by the β-adrenergic agonist isoprenaline (Fig. 1d; 1 μM, 5 min). This light response was significantly lower after pharmacologically blocking the adenylate cyclase with MDL-12,330A (Fig. 1d), which is in accordance with Bailes et al. who showed blunted JellyOp effects by $G_s$ but not by $G_q$ or $G_i$ inhibition[9]. Importantly, EBs expressing only GFP under control of the same promoter showed no light response and had identical basal cAMP levels compared to JellyOp EBs kept in the dark (Fig. 1d), excluding side effects or dark activity of JellyOp. Application of light with increasing intensities to purified ESC-derived cardiomyocytes induced a gradual increment of cAMP levels with a sigmoidal relationship on the logarithm of light intensity (Fig. 1e). Although we cannot fully exclude the activation of other G-proteins and intracellular pathways (Arrestin, CaMKII), our results clearly show that optogenetic stimulation with JellyOp enables fine control over cAMP levels in cardiomyocytes.

Spontaneously beating EBs were used to characterize the positive chronotropic response upon light stimulation. Baseline beating frequency in the dark was similar in JellyOp and GFP control EBs (Fig. 2a) suggesting the lack of JellyOp dark activity which would affect basal pacemaker activity. Brief illuminations (309 nW mm$^{-2}$, 20 s) of contracting areas within JellyOp EBs caused an instantaneous increase in beating frequency (Fig. 2b, Supplementary Movie 1), which was not seen in GFP-expressing control EBs. Light stimulation of JellyOp EBs induced an increase in beating rate which was similar to stimulation with isoprenaline

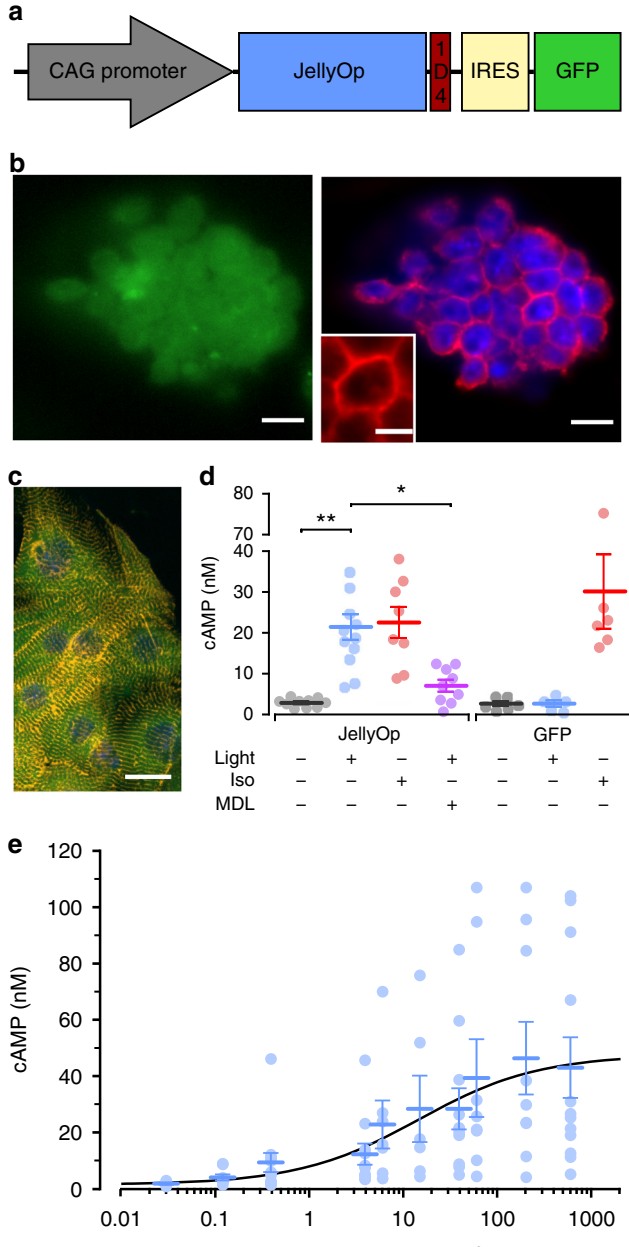

**Fig. 1** Generation of JellyOp expressing mouse ESCs and light-induced cAMP production in cardiomyocytes. **a** Plasmid for expression of JellyOp in fusion with the 1D4 rhodopsin epitope and with an internal ribosome entry site (IRES) for co-expression of the green fluorescence protein (GFP) under control of the chicken β-actin promoter (CAG). **b** Immunostaining of a transgenic ESC colony expressing GFP (green) and JellyOp (red: 1D4 rhodopsin epitope staining) (nuclear staining in blue, bars: 10 μm; insert: 5 μm). **c** GFP positive (green) cardiomyocytes indicated by α-actinin (yellow) staining (nuclear staining in blue, bar: 20 μm). **d** cAMP levels in JellyOp and GFP EBs after illumination (2.9 mW mm$^{-2}$, 5 min, MDL: 100 μM MDL-12,330A) or isoprenaline (Iso, 1 μM, 5 min) application ($n = 5$–12, Welch ANOVA: $p = 0.0014$, Games–Howell post-test: *$p < 0.05$, **$p < 0.01$). **e** Relationship between cAMP levels and light intensity in lactate-purified (see Methods) cardiomyocytes fitted with Hill equation ($n = 5$–12). Error bars: S.E.M.

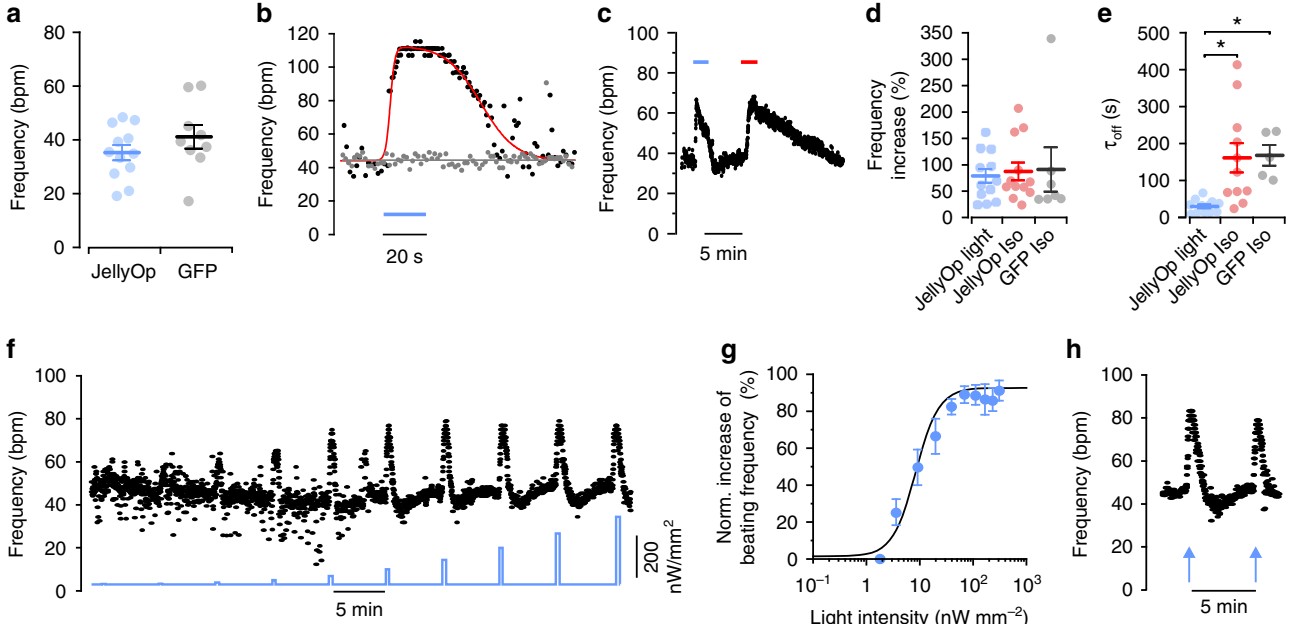

**Fig. 2** JellyOp activates $G_s$-signaling in ESC-derived cardiomyocytes. **a** Baseline frequencies of unstimulated spontaneously beating EBs expressing JellyOp ($n = 12$) or GFP only ($n = 9$) (two-sided unpaired Student's $t$-test: $p = 0.2527$). **b** Spontaneous beating frequency during illumination (blue bar, 309 nW mm$^{-2}$) of a JellyOp (black dots, fit in red see Methods) and a GFP control EB (gray). **c** Representative frequency trace of a JellyOp EB during stimulation by light (blue bar, 309 nW mm$^{-2}$) or with isoprenaline (red bar, 1 µM). **d**, **e** Statistical analysis of maximal beating rate increase (**d**) and deactivation time constant ($\tau_{off}$) after termination of stimulation (**e**) using JellyOp and GFP control EBs ($n = 5$–12, one-way ANOVA: $p = 0.924$ (**d**), Welch ANOVA, $p = 0.003$ (**e**), Games–Howell post-test: *$p < 0.05$). **f** Representative frequency trace of a JellyOp EB during stimulation by light pulses (20 s, 5 min delay) with stepwise increasing light intensities (2–309 nW mm$^{-2}$). **g** Relationship between normalized peak frequency increase and light intensity fitted with Hill equation ($n = 5$). **h** Frequency trace upon illumination with two supramaximal light intensities (blue arrows: left 1770 and right 2370 nW mm$^{-2}$, 20 s). Error bars: S.E.M.

in JellyOp and GFP control EBs (Fig. 2c, d). In contrast, return to baseline was significantly (~5 times) faster after termination of optogenetic stimulation compared to wash out of isoprenaline (Fig. 2c, e). This suggests that JellyOp expression does neither sensitize nor inhibit the native β-adrenergic signaling cascade and that optogenetic activation of $G_s$-signaling has identical effectiveness but higher temporal precision compared to agonist application.

To demonstrate fine graded control of beating rate, we stimulated EBs with light pulses (20 s) of stepwise increasing light intensities (Fig. 2f). This resulted in a gradual increase of beating frequency with a sigmoidal dependency on the logarithm of applied light intensity (Fig. 2g), similar to a dose–response relationship using a receptor agonist. The data points could be fitted with the Hill equation yielding a half maximal effective light intensity (ELi50) of 8.4 nW mm$^{-2}$ and a maximal effect at intensities of ~100 nW mm$^{-2}$. Supramaximal stimulation with up to 10-fold higher light intensities did not enhance the effect nor did it show desensitization, when applied repetitively with 300 s intervals between stimulations (Fig. 2h).

Next, we tested if graded control of $G_s$-signaling could be also achieved in EBs by varying the duration of JellyOp stimulation using light pulses between 1 ms and 20 s at two different light intensities, which are both supramaximal at 20 s pulse duration. Again, we observed a sigmoidal dependence of the chronotropic response on the logarithm of pulse duration and the 50% effect was obtained with an exposure time of 6.2 ms or 415 ms using 30.9 µW mm$^{-2}$ or 0.309 µW mm$^{-2}$ light pulses, respectively (Fig. 3a). To quantitatively compare these values, we calculated the amount of applied photons (quantum density) during each light pulse and found almost identical photon response-curves for both light intensities (Fig. 3b). Thus, activation of the $G_s$-pathway

with JellyOp is proportional to the product of light intensity and pulse duration and both parameters can be used to titrate the cellular response.

Besides brief light pulses, we also tested the response to prolonged illuminations over 10 min. This induced an instantaneous acceleration of EB beating frequencies to a maximum of $71.4 \pm 5.3\%$ increase (mean ± S.E.M., $n = 16$) and a subsequent inactivation with decline of chronotropy to a steady state rate ($19.8 \pm 2.9\%$ increase of baseline frequency) with a time constant of $93.4 \pm 14.0$ s (Fig. 3c). Interestingly, after 10 min continuous stimulation, the cardiomyocytes had to recover for at least ~5 min in order to induce a similar frequency increase (>90% of the first stimulation, Fig. 3d, blue dots). In contrast, recovery rate was much faster after 20 s stimulations and >90% recovery was reached already after 20 s waiting (Fig. 3d, black dots).

Thus, JellyOp can be used to stimulate $G_s$-signaling in cardiomyocytes in vitro repetitively without strong desensitization, when using brief (<20 s) illuminations. Because short and repetitive stimulations would be technically very challenging or even impossible using receptor agonists, our approach will enable experimental paradigms to explore the differential impact of long-lasting vs. pulsatile $G_s$-activation patterns to examine the effects of refractoriness, recovery and desensitization of G-proteins and their downstream effectors.

## Optogenetic $G_S$-signaling in cardiomyocytes from JellyOp mice. To analyze JellyOp function in the intact heart, we generated a transgenic JellyOp mouse line from the JellyOp G4 ESCs, which showed bright GFP expression in the atria and ventricles of the heart (Fig. 4a). Histological sections of the heart showed cytosolic GFP expression and membrane-bound JellyOp signal in

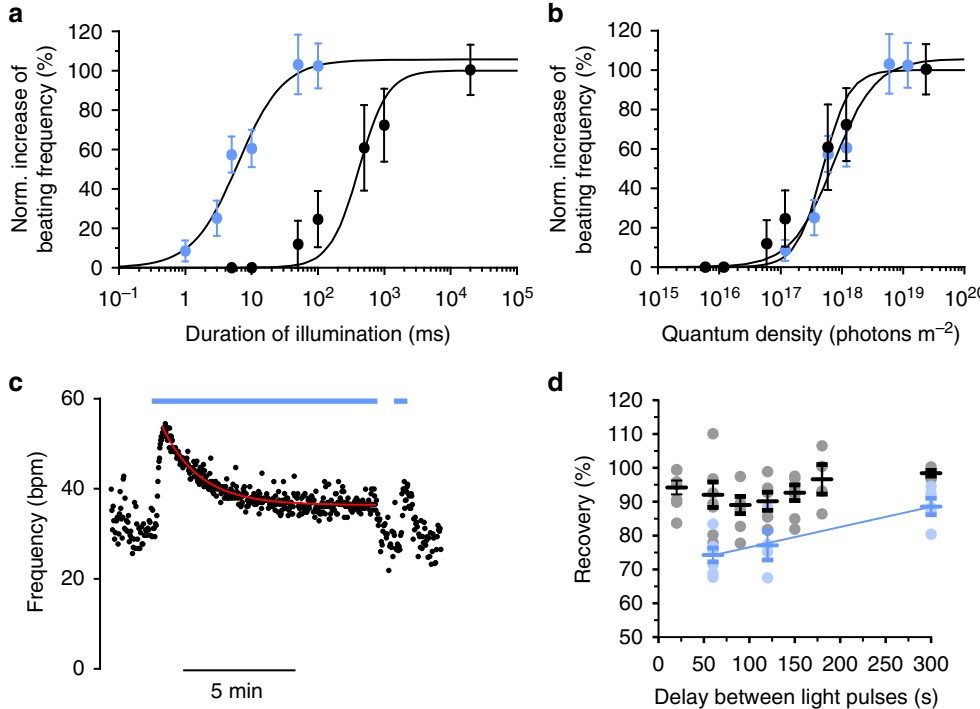

**Fig. 3** Analysis of JellyOp light sensitivity and recovery in ESC-derived cardiomyocytes in vitro. **a**, **b** Relationship between normalized peak frequency increase and duration of light pulses (**a**) or applied photon-density (**b**) for light pulses with 30.9 (blue dots) and 0.309 (black dots) µW mm$^{-2}$ ($n = 6$). **c** Representative frequency trace with long illumination (blue bar, 309 nW mm$^{-2}$, 10 min, inactivation fit in red) and a subsequent second light pulse (20 s). **d** Percentage of recovery of frequency response after a short (20 s, black dots) or a long-lasting (10 min, blue dots) illumination (309 nW mm$^{-2}$) analyzed with a second light pulse at various delay after the first light pulse. Error bars: S.E.M.

all ventricular cardiomyocytes (Fig. 4b). It is well known that chronic β-adrenergic stimulation of the heart causes alterations in Ca$^{2+}$-handling and cardiac hypertrophy with subsequent heart failure[10,11]. JellyOp mice were viable and did not show any obvious phenotype. To exclude possible hypertrophic side effects, we determined the heart weight to femur length ratio and could not detect a difference between JellyOp and wild-type mice (Fig. 4c). Ventricular cardiomyocytes were dissociated and investigated using patch-clamp and contraction measurements. Illumination with blue light (100 µW mm$^{-2}$, 10 s) resulted in an instantaneous increase of L-type Ca$^{2+}$ current ($I_{Ca,L}$), which reached a maximal increase of 20.5 ± 5.4% within 63.8 ± 7.9 s after the start of illumination (Fig. 4d–f). This response was smaller compared to the effect of β-adrenergic stimulation with iso-prenaline (1 µM) that led to an $I_{Ca,L}$ increase of 43.3 ± 6.7% (Fig. 4g). Contractility and relaxation speed were analyzed by edge detection of isolated, electrically paced ventricular cardio-myocytes. Similarly to the effects on $I_{Ca,L}$, illumination (100 µW mm$^{-2}$, 10 s) led to an instantaneous enhancement of the degree of contraction with a maximal increase of 218 ± 70% and faster relaxation with a reduction of time constant $\tau_{rel}$ by 38.6 ± 6.5% (Fig. 4h–l). The smaller stimulatory effect on $I_{Ca,L}$ (+21%) compared to the effect on contractility (+218%) is most likely due to the fact that cAMP-dependent PKA phosphorylates, besides the L-type Ca$^{2+}$-channels, also other target proteins involved in the regulation of contractile force, such as the ryanodine-receptor type 2 and phospholamban and thereby augments function of the sarco/endoplasmic reticulum Ca$^{2+}$-ATPase, which altogether increases the Ca$^{2+}$-induced Ca$^{2+}$ release mechanism. Interest-ingly, maximal relaxation speed was reached 46.3 ± 12.7 s after the onset of illumination, which was significantly faster than the maximal effect on contraction, which was observed after 92.9 ± 14.4 s (Fig. 4m). This might indicate privileged phosphorylation

of the phospholamban microdomain compared to bulk cytosolic phosphorylation of L-type Ca$^{2+}$ channels and ryanodine-recep-tors, as reported earlier[12]. Interestingly, the fast effect on relaxation speed after β-adrenergic stimulation is well in line with previously reported phospholamban phosphorylation data obtained from isolated ventricular mouse cardiomyocytes[13] or freeze clamped rat hearts[14].

**Spatial difference in G$_s$-signaling in the atria.** To modulate chronotropy of the heart, we investigated the atria of JellyOp mice and found bright GFP fluorescence and membrane-bound JellyOp signals in atrial cardiomyocytes (Fig. 5a). Illumination (2.9 mW mm$^{-2}$, 5 min) of isolated atrial tissue significantly increased cAMP levels by 224%, which was similar to the application of 1 µM isoprenaline (Fig. 5b). The ability of JellyOp to modulate the beating rate was analyzed in explanted, Langendorff-perfused hearts. Supramaximal illumination (2 mW mm$^{-2}$, 90 s) of the right atrium, in which the sinus node is located, led to a maximal acceleration of heart rate by 44.0 ± 4.1% (Fig. 5c, e). After reaching the peak frequency, hearts showed a clear reduction of the chronotropic response during illumination with a time constant of 33.5 ± 18.1 s to a steady state level of 28.8 ± 8.5% ($n = 5$) heart rate acceleration. This time course and the decline to a steady state level of 65% of the peak effect is very similar to that of cAMP levels in isolated ventricular mouse cardiomyocytes during con-tinuous isoprenaline stimulation[15] and was explained in a math-ematical model of β-adrenergic signaling by phosphodiesterase activity, cAMP fluxes between compartments, and receptor desensitization[16]. Compared to illumination, pharmacological stimulation with 1 µM isoprenaline resulted in slightly higher maximal heart rates (Fig. 5d, e), but only after much longer sti-mulation duration (Fig. 5d, f). Importantly, the chronotropic effect returned almost instantaneously to baseline after termination of

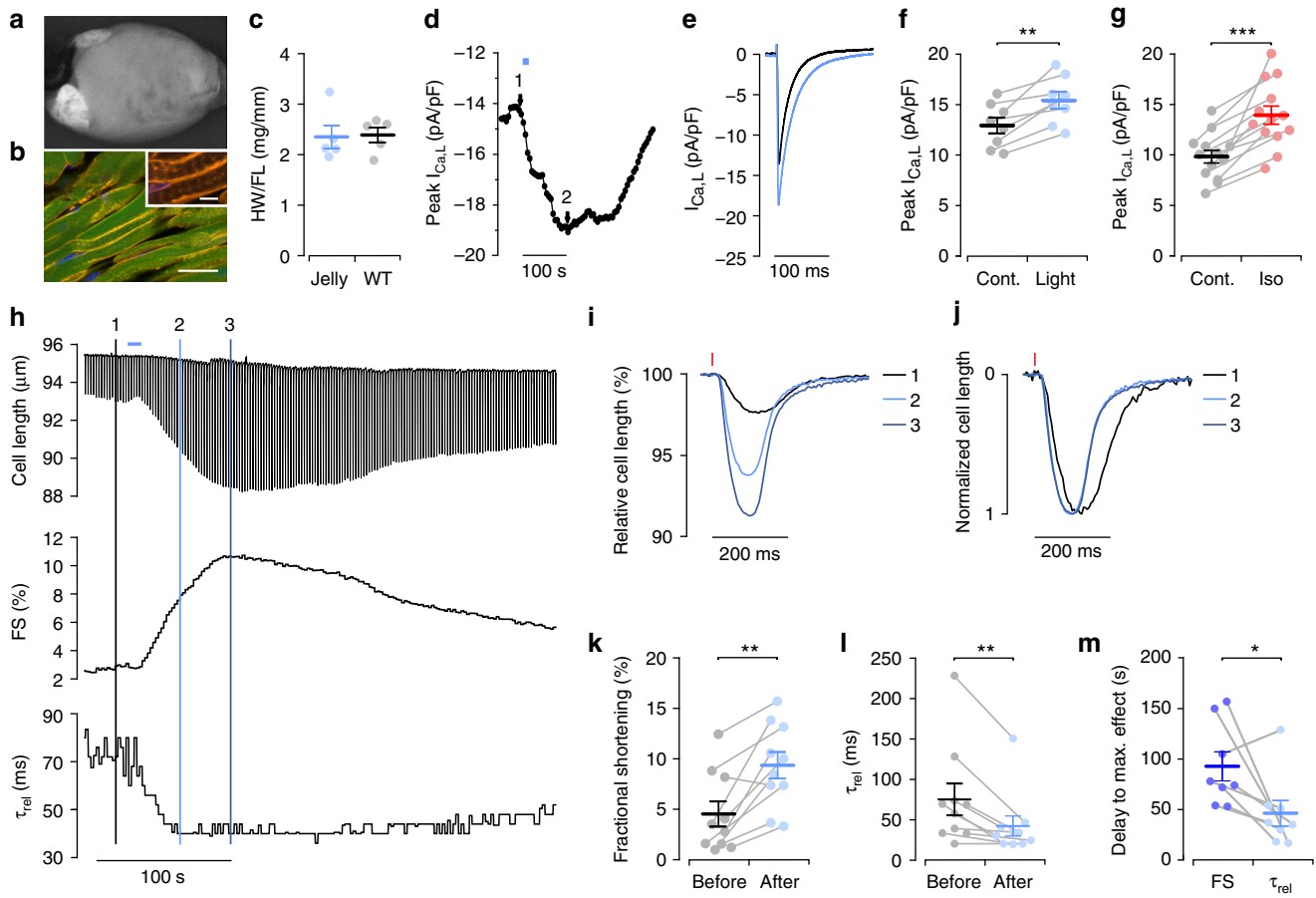

**Fig. 4** JellyOp function in isolated mouse ventricular cardiomyocytes. **a** Overview of GFP fluorescence in an intact JellyOp mouse heart. **b** Section of the left ventricle with membrane-bound JellyOp-1D4 epitope signals (yellow) in GFP (green) positive cardiomyocytes (nuclear staining in blue, bar: 20 μm, insert: 5 μm). **c** Heart weight (HW) to femur length (FL) ratio in JellyOp (Jelly) and wild-type (WT) mice (>6 months of age, $n = 5$, two-sided unpaired Student's $t$-test: $p = 0.8915$). **d, e** Representative changes in peak L-type $Ca^{2+}$ current ($I_{Ca,L}$) in a JellyOp ventricular cardiomyocyte upon illumination (**d**, blue bar, 100 μW mm$^{-2}$, 10 s) and original traces taken from time points indicated (**e**). **f, g** Statistical analysis of maximal $I_{Ca,L}$ before (Contr., black) and after illumination (**f**, blue, 100 μW mm$^{-2}$, 10 s, $n = 8$, two-sided paired Student's $t$-test: $p = 0.0035$) or isoprenaline application (**g**, red, 1 μM, 75–100 s, $n = 13$, two-sided paired Student's $t$-test: $p = 0.00003$). **h** Original trace (upper panel) of cell length measurement from an electrically stimulated isolated JellyOp ventricular cardiomyocyte upon illumination (blue bar, 100 μW mm$^{-2}$, 10 s) with corresponding fractional shortening (FS, middle panel) and relaxation constant $\tau_{rel}$ (lower panel). **i, j** Single traces of relative cell length in % of resting length (**i**) and normalized to maximal shortening (**j**) evoked by electrical stimulation (red line) before (1, black), 40 s (2, light blue) and 80 s (3, dark blue) after illumination (time points indicated in (**h**)). **k, l** Statistical analysis of maximal fractional shortening (**k**) and relaxation speed ($\tau_{rel}$, **l**) before (black) and after (blue) illumination (100 μW mm$^{-2}$, 10 s, $n = 10$, two-sided paired Student's $t$-test, $p = 0.0011$ (**k**); Wilcoxon signed rank test, $p = 0.004$ (**l**)). **m** Delay from illumination onset to maximal effect on fractional cell shortening (FS, dark blue) and relaxation speed ($\tau_{rel}$, light blue) ($n = 8$, Wilcoxon signed rank test, $p = 0.04$). Error bars: S.E.M.

illumination in contrast to a sustained effect after wash out of isoprenaline (Fig. 5g). The spectral sensitivity of JellyOp was tested by illuminating the right atrium with monochromatic light between 400 and 700 nm, which resulted in a frequency increase between 400 and 600 nm (Fig. 5h). These data were fitted with the template of an opsin retinaldehyde pigment (Govardovskii nomogram)[17] resulting in a peak wavelength ($\lambda_{max}$) of 493 nm (Fig. 5h, red line), which is in accordance with the reported action spectrum of wild-type JellyOp[18]. This data and the fit (Eq. (2), see Methods) can be used to estimate the required light intensity at less optimal wavelengths, e.g., to obtain the effect of 493 nm (10 μW mm$^{-2}$) ~800 μW mm$^{-2}$ would be required at 600 nm. However, we would like to point out that light penetration into the tissue surrounding the sinus node is wavelength dependent and light can be toxic at high intensities (e.g., calculated 51 mW mm$^{-2}$ at 650 nm is unrealistic).

To analyze the delay of $G_s$-induced heart rate modulation, we applied very brief light pulses (1 mW mm$^{-2}$, 100 ms) to the right atrium. This led to an almost instantaneous acceleration of the spontaneous sinus rhythm by 14.2% starting already ~300 ms after the onset of illumination (Fig. 6a) and reaching a maximum frequency after $17.8 \pm 3.3$ s ($n = 7$). Application of light pulses (100 ms) with various intensities resulted in an ELi50 for light-induced heart rate increase of 386 μW mm$^{-2}$ (Fig. 6b). The 5-fold difference in temporal kinetics of sinus rate acceleration (peak in <20 s) compared to the PKA effects in isolated ventricular cells (peak in ~100 s) is most likely caused by the direct action of cAMP on the pacemaker HCN-channels in sinus node cells[19]. Importantly, the very short delay from light to rate response correlates well with recent reports using optogenetic stimulation of cardiac sympathetic neurons[20,21].

It is known that atrial premature contractions inducing atrial fibrillation are often originating from the ostia of pulmonary veins in the dorsal left atrium[22,23]. To prove that localized β-adrenergic effects can be investigated using our optogenetic method, we applied light pulses to the dorsal

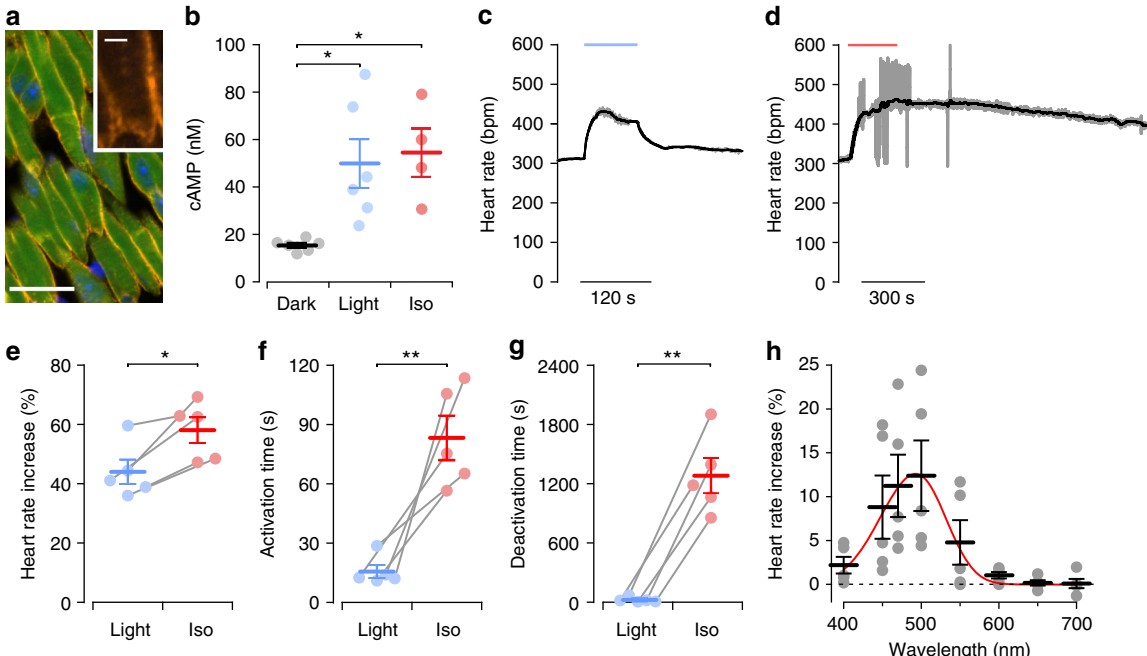

**Fig. 5** JellyOp function in mouse atria of intact hearts. **a** Section through the right atrium shows GFP (green) and membrane-bound JellyOp-1D4 epitope signals (yellow, nuclear staining in blue, bar: 20 μm, insert: 5 μm). **b** Analysis of cAMP levels in atrial tissue from JellyOp mice in the dark and after illumination (2.86 μW mm$^{-2}$, 5 min) or stimulation with isoprenaline (Iso, 1 μM, 5 min) ($n = 4$–6, one-way ANOVA: $p = 0.0074$, Bonferroni post-test: *$p < 0.05$). **c**, **d** Representative heart rate traces of a JellyOp heart during maximal light stimulation of the dorsal right atrium (**c**, blue bar, 2 mW mm$^{-2}$, 90 s) or during perfusion with isoprenaline (**d**, red bar; 1 μM, 4 min) (gray: original trace, black: running average values). **e**–**g** Statistical analysis of the maximal relative heart rate increase (**e**), activation time (**f**, stimulation start to 80% of maximum rate), and deactivation time (**g**, stimulation end to 50% of maximum rate) after illumination (blue, 2 mW mm$^{-2}$, 90 s) and perfusion with isoprenaline (red, 1 μM, 4 min) ($n = 5$, two-sided paired Student's $t$-test: $p = 0.0248$ (**e**), $p = 0.0058$ (**f**), $p = 0.0017$ (**g**)). **h** Relative heart rate increase after illumination of the dorsal right atrium with light of wavelengths of 470 nm and 400–700 nm in 50 nm steps (10 μW mm$^{-2}$, 1 s, $n = 5$) fitted with the Govardovskii equation (red line, $R^2 = 0.98$). Error bars: S.E.M.

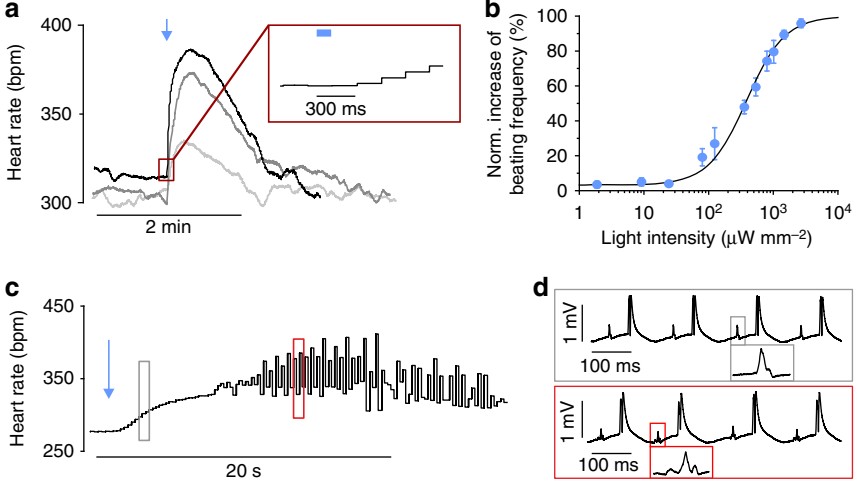

**Fig. 6** Spatial difference of JellyOp stimulation in the left and right atria. **a** Representative heart rate traces upon brief (blue arrow, 100 ms) illumination of the anterior right atrium with 80 (gray), 540 (dark gray), and 1015 (black) μW mm$^{-2}$. Insert (red box) highlights the short delay between light (blue bar) and start of heart rate acceleration. **b** Statistical analysis of experiments described in (**a**) showing the relationship between normalized peak heart rate increase and light intensity fitted with Hill equation ($n = 7$). **c** Representative heart rate trace upon illumination of the dorsal left atrium (blue arrow, 190 μW mm$^{-2}$, 1 s) resulting in irregular beating. **d** ECG traces from the experiment shown in (**c**) before (gray box) and during a period with premature atrial contractions (red box) indicated by altered $P$-wave morphology (see inserts). Error bars: S.E.M.

left atrium (190 μW mm$^{-2}$, 1 s). This initially induced a mild acceleration of sinus rhythm (possibly due to scattered light to the sinus node), which was subsequently interposed by spontaneous premature atrial contractions (Fig. 6c). These were characterized by premature P-waves with different morphology compared to

regular sinus rhythm (Fig. 6d) indicating an origin outside the sinus node, most likely from the illuminated region. Importantly, we have not observed such premature atrial contractions when flashing light with identical parameters on the right atrium of JellyOp mice.

Because of the lack of cell-type specific expression of JellyOp, definitive conclusions on the mechanism and the cell types involved in the observed supraventricular arrhythmia cannot be drawn at this stage. In the future, cell-type specific expression of JellyOp using Cre/LoxP systems in cardiomyocytes, fibroblasts, endothelial- and smooth muscle cells will allow the analysis of direct and paracrine effects of β-adrenergic signaling in the different cell types of the heart.

Taken together, we herein present JellyOp as an optogenetic tool to investigate $G_s$-signaling in the heart. The precise spatial and temporal control allows novel stimulation protocols to determine the physiological and pathophysiological effects of $G_s$-activation for pacemaker function and arrhythmia generation in the intact heart. As a proof-of-concept, we have investigated the precise temporal delay between activation of the receptor and downstream effects on pacemaker activity and contractility in vitro as well as in the intact heart and demonstrated spatial differences of $G_s$-signaling in the left and right atria. Importantly, we could not detect any side effects on the heart by over-expressing JellyOp indicating a lack of adverse dark activity.

When comparing JellyOp stimulation to pharmacological stimulation of β-receptors with isoprenaline, similar effects on beating rate increase of ESC-derived cardiomyocytes were observed (light: 79%, isoprenaline: 87%, Fig. 2d), but the effect of light was significantly lower on $I_{Ca,L}$ in ventricular cardiomyocytes (light: 21%, isoprenaline: 43%) or heart rate of sinus node cells in the intact heart (light: 44%, isoprenaline: 58%, Fig. 5e). At this stage, we cannot exclude that JellyOp slightly also activates $G_i$-proteins, and thereby impairs the $G_s$ effect. However, this could also be explained by privileged signaling of β-receptors to the $G_s$-proteins and adenylate cyclases in microdomains compared to an overexpressed receptor from the lower invertebrate prebilaterian animal Jellyfish. Targeting JellyOp to specific compartments by transplanting arrestin or phosphodiesterase interacting protein sequences[24] from β1 or β2-receptors to JellyOp will be a very powerful tool to determine the influence of macromolecular signaling complexes on β-adrenergic signaling with high temporal resolution.

In general, direct optogenetic stimulation of G-protein coupled receptors will provide new insights into temporal and spatial aspects of G-protein and downstream signaling, and the consequences for cardiovascular (patho-)physiology.

## Methods

**Generation of the JellyOp IRES GFP construct.** The CAG-JellyOp-IRES-GFP expression vector was designed by excising the JellyOp-1D4 epitope fragment from the 1256pcDNA3.1-JellyOp plasmid (kindly provided by R. Lucas, University of Manchester) by XhoI and SacI and cloning it into the CAG-mOPN4-IRES-GFP vector[7] from which mOPN4 had been removed by BamHI/SacI excision. The plasmid was confirmed by sequencing and linearized with BglII before transfection.

**ESC culture, transfection, and differentiation.** G4 hybrid ESCs (mycoplasma negative, kindly provided by A. Nagy and M. Gertsenstein, Mount Sinai Hospital, Toronto) were cultured on neomycin-resistant mouse embryonic fibroblasts in high-glucose Dulbecco's Modified Eagle's Medium (KnockOut DMEM, Invitrogen) supplemented with 15% FCS (PAN-Biotech), 0.1 mM nonessential amino acids, 100 U mL$^{-1}$ penicillin, 100 mg mL$^{-1}$ streptomycin, 2 mg mL$^{-1}$ L-glutamine (all Invitrogen), 0.1 mM β-mercaptoethanol (Sigma-Aldrich), and 500 U mL$^{-1}$ leukemia inhibitory factor (Chemicon). DNA transfection was performed by electroporation of $4 \times 10^6$ ES cells in PBS (Invitrogen) mixed with 40 μg of the linearized CAG-JellyOp-IRES-GFP plasmid with a single electrical pulse (250 V, 500 μF, Bio-Rad Gene Pulser)[7,25]. Electroporated ESCs were plated and 24 h after transfection, 165 μg mL$^{-1}$ G418 was added to the medium to select for neomycin-resistant colonies. GFP positive clones were picked and further propagated. ESCs with stable expression of GFP under control of the CAG promoter served as controls[7,25]. During ESC culture, 9-Cis retinal was not supplemented. ESCs were differentiated within EBs using the hanging drop method (20,000 cells per drop)[7,25]. After 2 days, hanging drops were washed and kept in suspension on a horizontal shaker for 3 days. At day 5 of differentiation, EBs were either plated on 0.1% gelatin-coated

glass coverslips for analysis of beating frequency and cAMP concentrations or used for cardiomyocyte purification (see below).

**cAMP measurements.** Light-induced cAMP production in EBs compared to pharmacological stimulation (Fig. 1d) was performed on day 10 of EBs differentiation. For a detailed analysis of light sensitivity of cAMP production (Fig. 1e), cardiomyocytes were purified using lactate replacement of glucose as reported earlier[26] with modifications. Briefly, EBs with clusters of spontaneously beating cardiomyocytes were cultured in high-lactate, non-glucose IMDM from day 8 to day 13 and subsequently dissociated on day 14 in 15 mL falcon tubes with collagenase 4 (280 U mL$^{-1}$, Worthington Biochemical Corporation) on a thermomixer (10 min, 900 rpm, 37 °C). Dissociated cells were replated on 10 μg mL$^{-1}$ fibronectin-coated cell culture plates and cultured in normal IMDM medium +20% FCS for 1 day. The following 5 days, cells were kept in high-lactate IMDM and on day 20, purified cardiomyocytes were dissociated for a second time and 5000 cells were plated per well of a 24 multiwell plate. EBs or purified cardiomyocytes were supplemented with 9-Cis retinal (1 μM, Sigma-Aldrich) for 30 min and 3-isobutyl-1-methylxanthine (IBMX, 50 μM, Sigma-Aldrich) directly before stimulation.

For measurement of cAMP in atrial tissue, left and right atria from JellyOp mice were harvested and immobilized by minutien pins in a silicone-coated 3 cm culture dish with 1 mL Tryode solution containing 1 μM 9-Cis retinal and 100 μM IBMX. Light stimulation (2.9 mW mm$^{-2}$, 5 min) was performed using a macroscope (MVX10, Olympus) with a ×1 objective (MVPLAPO1x), a fluorescence microscope light source (X-Cite 120PC, Lumen Dynamics) and a 475/64 bandpass filter (AHF Analysetechnik). After illumination, cells were incubated for 1 h at 37 °C before cell lysis and atrial tissue was manually dissected and dissolved in lysis buffer for 2 h in a thermomixer (21 °C and 900 bpm). cAMP levels were measured in the supernatant after lysis with a homogeneous time-resolved fluorescence competitive immunoassay (cAMP dynamic 2 kit, Cisbio) and a plate reader (Infinite F200 PRO, Tecan) according to the manufacturer's instructions.

**Frequency analysis of ESC-derived cardiomyocytes.** EBs were plated at day 5 of differentiation on 0.1% gelatin-coated glass coverslips and spontaneous beating was analyzed at day 10. Thirty minutes before the experiment, medium was replaced by Tyrode solution (in mM: 142 NaCl, 5.4 KCl, 1.8 CaCl$_2$, 2 MgCl$_2$, 10 glucose, and 10 HEPES; pH 7.4) supplemented with 1 μM 9-Cis retinal (Sigma-Aldrich). During experiments, EBs were constantly superfused with Tyrode solution without 9-Cis retinal at ~35 °C.

Light stimulation of spontaneously contracting EBs was performed as reported before[7] with a temperature controlled LED module (Omicron LEDMOD LAB, 470 nm, Omicron Laserage) operated by a signal generator (Model 2100, A-M Systems) and coupled to an Axiovert 200 microscope (Zeiss) with an optical fiber. Light was attenuated with a 1% neutral density filter and directed to a ×20 Fluar objective (numerical aperture: 0.75, Zeiss) by a 580 nm dicrotic filter (AHF Analysetechnik). Light intensity was determined at the objective with a power meter and appropriate wavelength correction (PM100 powermeter and S130A sensor, Thorlabs).

To avoid JellyOp activation, an infrared LED (760 nm, 11.8 μW mm$^{-2}$ at the focal plane) was used for imaging. Spontaneous beating was recorded with a CCD camera (piA640-210gm, Basler) at 51 frames per second and analyzed online using custom-designed software (LabView, National Instruments) as described before[7]. Experiments with frequencies >270 bpm were excluded from analysis because of false detection problems. Only EBs with stable beating frequency (regular baseline beating over 2 min without interruption or arhythmical episodes) before the start of illumination were used for statistical analysis. To reduce experimental noise from single arrhythmic contractions, the raw frequency response to each light pulse was fitted with an Asym2Sig fit formula (Origin 8, OriginLab)

$$y = y_0 + A \frac{1}{1 + e^{-\frac{x - x_c + w_1/2}{w_2}}} \left(1 - \frac{1}{1 + e^{\frac{x - x_c + w_1/2}{w_3}}}\right) \quad (1)$$

and the resulting fit trace (Fig. 2b, red line) was used to determine the peak frequency and the mean baseline frequency during 60 s before stimulation. Time constants for inactivation from peak to plateau frequency and for deactivation from peak to baseline frequency were determined by an exponential decay fit (ExpDec1, Origin 8).

**Generation of transgenic mice and heart weight measurements.** Animal experiments were performed in accordance with the guidelines from Directive 2010/63/EU of the European Parliament on the protection of animals used for scientific purposes and were approved by the local ethics review board (Az. 84-02.04.2012A146). Transgenic mice were generated by aggregation of JellyOp expressing, transgenic G4 ESCs with 40 chromosome karyotype and diploid morula stage CD-1 embryos as described previously[25]. Chimeric mice, as identified by their coat color chimerism, were bred to CD-1 mice to test germline transmission. Offspring with agouti coat color were analyzed for inheritance of the transgene by detection of GFP signal in tail tissue.

Hearts were explanted and dried for 5 h at 37 °C. Dry heart weight was normalized to the femur length. The age of mice was >6 months.

**Isolation of adult ventricular cardiomyocytes**. Ventricular cardiomyocytes were isolated from JellyOp mice, as described previously[25]. Briefly, hearts were perfused in the Langendorff configuration with dissociation solution (in mM: 135 NaCl, 4 KCl, 1 MgCl$_2$, 2.5 HEPES, 5 glucose, 25 butanedione monoxime; pH 7.4) for 5 min at 37 °C and then 50 µM CaCl$_2$, 0.8 mg mL$^{-1}$ collagenase B (Roche) and 0.3 mg mL$^{-1}$ trypsin (Invitrogen) was added for 12–13 min. Ventricles were cut in small pieces and mechanically dissected, cells were filtered through a nylon mesh and the pellet was resuspended in dissociation solution with 50 µM CaCl$_2$, 5% FCS but without butanedione monoxime. Subsequently Ca$^{2+}$ was increased in four steps to 1.8 mM over 40 min and finally 1 µM 9-Cis retinal was added for at least 30 min before patch clamp experiments.

**Electrophysiological measurements**. Single cells were plated at low density on laminin-coated (0.1%) coverslips in $I_{Ca,L}$ external solution containing (in mM) 120 NaCl, 5 KCl, 1.8 CaCl$_2$, 1 MgCl$_2$, 10 HEPES, 20 TEACl, pH 7.4 with TEAOH. Patch-clamp experiments were performed using an EPC10 amplifier (Heka) in the whole cell configuration. $I_{Ca,L}$ was recorded in the voltage clamp mode, the pipette solution contained (in mM) 120 CsCl, 3 MgCl$_2$, 3 NaATP, 0.42 NaGTP, 5 Na Phosphocreatine, 10 EGTA, and 5 HEPES, pH 7.2 with CsOH. For measuring peak $I_{Ca,L}$ cardiomyocytes were held at a holding potential of −80 mV and a 50 ms long depolarizing voltage step to −40 mV was applied to inactivate $I_{Na}$, followed by a 300 ms depolarizing voltage step to 10 mV to evoke $I_{Ca,L}$ every 5 s. Data were acquired at a sampling rate of 20 kHz, filtered at 1 kHz, digitized and analyzed with the Patchmaster and Fitmaster software (Heka).

**Contractility measurements of ventricular cardiomyocytes**. After heart dissociation, a part of ventricular cardiomyocytes was kept at 4 °C in suspension in Tyrode solution containing 1 µM 9-Cis retinal, plated on the next day at low density on laminin-coated (0.1%) coverslips and superfused with Tyrode solution at ~35 °C. Cardiomyocytes were electrically (2 ms biphasic pulses, 40 V) paced at 0.5 Hz with two platinum electrodes and a stimulator (Myotronic). Contractions of single cardiomyocytes were recorded with an edge detection system (Myocam S and IonWizard software, Ionoptix). Data was recorded and analyzed with a Powerlab system and the LabChart software (AD Instruments). The average cell shortening of 11 contractions before illumination was compared with the average cell shortening of 11 beats around the maximal peak of cell shortening. Relaxation kinetics were determined by exponential decay fit from 85 to 15% of peak height.

**Stimulation of explanted hearts ex vivo**. Mice were heparinized and sacrificed by cervical dislocation. Hearts were explanted and perfused in Langendorff configuration with Tyrode solution. A bipolar cardiac electrogram was recorded with a silver-chloride electrode placed at the right atrium and a metal spoon under the apex of the heart using a bio-amplifier recording system (PowerLab 8/30, Animal Bio Amp ML 136, LabChart 7.1 software, AD Instruments). Illumination was performed with a macroscope (MVX10, Olympus) equipped with a ×1 objective (MVPLAPO1x) using 470 nm LEDs (LEDC5 and LEDD1, Thorlabs or GCS-0470-50-A510 LED and BLS-13000-1, Mightex) attached to the epifluorescence port. The spatial extent of the illumination was varied using the zoom function of the macroscope. Light intensity was calibrated by with a powermeter (PM100 and S130A, Thorlabs).

To determine maximal frequency increase during light stimulation (Fig. 5c–g), the dorsal part of the right atrium was illuminated with supramaximal intensity (2 mW mm$^{-2}$, 90 s, 104 mm$^2$ area). After the return of the frequency to baseline, the heart was perfused with isoprenaline (1 µM, 4 min). Spectral sensitivity of JellyOp (Fig. 5h) was determined using a monochromator (OptoScan, Cairn-Research) to generate 1 s long illuminations with wavelengths of 470 nm and 400–700 nm in 50 nm steps. To determine peak wavelength ($\lambda_{max}$), normalized frequency values ($y$) at applied wavelength $\lambda$ were fitted with the Govardovskii normogram equation[17]:

$$y = 1/\{exp[A*(a-x)] + exp[B*(b-x)] + exp[C*(c-x)] + D\} \quad (2)$$

with $x = \frac{\lambda_{max}}{\lambda}$, $A = 69.7$, $a = 0.88$, $B = 28$, $b = 0.922$, $C = -14.9$, $c = 1.104$, $D = 0.674$

Frequency traces were smoothed with a triangular filter (Bartlett, 10 s window). The relative frequency response was normalized to baseline, which was defined as maximal frequency of the smoothed data during a 45 s time interval before stimulation.

To analyze light sensitivity (Fig. 6a, b), hearts were illuminated on the anterior right atrium with brief 100 ms 470 nm light pulses with stepwise increasing light intensities (2–2600 µW mm$^{-2}$). For induction of supraventricular extra beats (Fig. 6c, d), the dorsal left atrium at the region of pulmonary vein insertion was illuminated with 190 µW mm$^{-2}$ for 1 s.

**Histology and immunofluorescence staining**. Cells and hearts were fixated with 4% formaldehyde (PanReac AppliChem). ESCs were incubated with primary antibody against the 1D4 rhodopsin-epitope (1:100, StressMarq) diluted in 5% donkey serum and 0.2% Triton X (PanReac AppliChem) for 2 h at room temperature. Conjugated Alexa Fluor 647 secondary antibody (1:400, Invitrogen)

diluted in 1 µg mL$^{-1}$ Hoechst 33342 (Sigma-Aldrich) was applied for 1 h at room temperature. Images of ESCs were acquired using an inverted microscope (Eclipse Ti2, Nikon; Prime BSI camera, Photometrics; Micro-Manager software). 1D4 rhodopsin images were deconvoluted using the Richardson-Lucy Total Variation algorithm ($N = 20$, $\lambda = 0.0215$) and the Born and Wolf point spread function model (DeconvolutionLab2 and PSF Generator Plugins for ImageJ[27,28]).

Cryopreserved hearts were sectioned with a cryotome (Leica) into 10 µm thick slices. After permeabilization with 0.2% Triton X-100 (Fluka) for 10 min and blocking with 5% donkey serum (Jackson ImmunoResearch) for 20 min, primary antibodies against the 1D4 rhodopsin-epitope (1:400, StressMarq) or α-actinin (1:400, Sigma-Aldrich) diluted in 0.5% donkey serum were applied for 2 h at room temperature. Appropriate Cy-3 and Cy-5 conjugated secondary antibodies (1:400, Jackson ImmunoResearch) diluted in 1 µg mL$^{-1}$ Hoechst 33342 (Sigma-Aldrich) were applied for 1 h at room temperature. Fluorescence images of histological sections were acquired with an inverted microscope equipped with an optical section module (Axiovert 200 with ApoTome and AxioVision 4.2, Zeiss).

**Statistics**. Statistics were performed with GraphPad Prism 8.0, GraphPad Software. Data are shown as mean ± S.E.M. and a $p$-value < 0.05 was considered statistically significant. The $n$ values in the legends indicate the number of independent experiments (EBs, cardiomyocytes, atria, or hearts). Data were tested for normality (D'Agostino–Pearson test) and for equal variance (Brown–Forsythe test for ANOVA, $F$-test for unpaired Student's $t$-test). Non-normally distributed data in Fig. 4l, m were analyzed with the Wilcoxon signed-rank test for nonparametric data. Because of unequal variances, data in Figs. 1d, 2e were tested with Welch-ANOVA with Games–Howell post-test. All other data were tested with appropriate paired and unpaired Student's $t$-test or ANOVA analysis with Bonferroni post-test. Because of the exploratory nature of this study, effect sizes could not be predicted and thus prior power analysis to determine the sample sizes was not performed.

**Technical note on light sensitivity of JellyOp**. The light intensities required to activate JellyOp for in vitro assays are $10^4$–$10^5$ times lower than those needed for optogenetic stimulation of Channelrhodopsin2 in cardiomyocytes (~1 mW mm$^{-2}$)[25] or for stimulation of the artificial light-sensitive chimeric G$_s$-coupled receptor optoβ$_2$-AR (estimated from the cAMP assay in Airan et al.[29]). The high light sensitivity has advantages in regard to phototoxicity and penetration into tissue, but also results in some challenges for general handling of cells and performance of experiments. For instance, video microscopy had to be performed with infrared LED illumination (760 nm, 11.8 µW mm$^{-2}$) to obtain stable basal beating frequencies because white microscope light activated JellyOp resulting in an increase of basal beating frequency. Furthermore, the intensity of room light measured at the JellyOp-sensitive spectrum is in the range of 25 µW mm$^{-2}$. In our experimental setting partial activation of the G$_s$-signaling cascade by room light could be observed and therefore, all functional experiments were performed in dark rooms with dim red light illumination.

**Reporting summary**. Further information on experimental design is available in the Nature Research Reporting Summary linked to this article.

## Data availability
The data that support the findings of this study are available from the corresponding author upon reasonable request.

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

## Acknowledgements

The authors thank F. Holst and C.C. Vogt (University Bonn) for technical assistance, R. Lucas (Manchester) for providing the JellyOp plasmid and A. Nagy and M. Gertsenstein (Mount Sinai Hospital, Toronto) for providing the G4 mouse embryonic stem cell line. This work was funded by the Deutsche Forschungsgemeinschaft (DFG, German Research Foundation—SA1785/7-1, SA1785/8-1, SA1785/9-1, 214362475/GRK1873/2, Young Investigator Programme of the SPP1926 GO1011/11-1), and the Bonfor Program (Medical Faculty, University of Bonn).

## Author contributions

P.M., T. Bruegmann, B.K.F., and P.S. designed the study. P.M., T. Bruegmann, V.D., D. M., T. Beiert, and P.S. performed experiments and analyzed data. M.H. generated the transgenic mice.

## Additional information

**Competing interests:** The authors declare no competing interests.

