## [Peer Review File · Nature Communications]

Reviewers' comments:

Reviewer #1 (Remarks to the Author):

The manuscript reports application of a Gs-coupled opsin (JellyOp) to allow optogenetic control over cAMP concentration and contractile activity in ES cell derived cardiomyocytes and in the intact heart of a transgenic mouse lines. The approach is innovative and the data compelling. The obvious limitation in this dataset is that the technological breakthrough appears not to have been applied to achieve a great advance in understanding of physiology or as a step towards therapy.

Concentrating on the characterisation of the optogenetic tool, I found this comprehensive and convincing. JellyOp appears to function very well in this system and give an unprecedented level of control over cardiac activity. I have only a couple of suggestions. Firstly, The decay in heart rate over 5 min of light exposure is interesting but could have a number of origins. The most important is to distinguish whether it reflects decay of opsin activity vs cell response. I would therefore ask for a side by side comparison of a time course for application beta agonist and or iso. Do changes in heart rate also decay for those treatment? My second suggestion is that the authors explore the JellyOp driven response to longer wavelength light. Longer wavelengths penetrate biological tissue more effectively and it would be interesting to know if JellyOp can be activated by them at reasonable intensities.

Reviewer #2 (Remarks to the Author):

This is an interesting manuscript describing a nice way to potential understand Gs signalling in the heart using an optogenetic approach. The authors have developed a system using a novel opsin that can be used to active Gs signalling in cells that express it when stimulated with light. They demonstrate clearly that it activates Gs signalling in cardiomyocytes and in hearts from ES-derived cells and transgenic animals. The experiments are in general well designed and well performed. Some concerns that could addressed:

1) the authors do not demonstrate or know whether JellyOp activates other G proteins. It would be important to tease this out, if they want to claim that it is specific for Gs signalling. Is this an issue for comparing the effects of isoproterenol with JellyOp activation?

2) Why aren't the kinetics of relaxation affected in muscle cells? This is surprising given that SERCA2 is a target of PKA.

3) A better transgenic model might have been one that is cardiomyocyte specific. Given that there is limited data associated with arrhythmias in Figure 2 perhaps the connection to disease should be tempered?

Minor issues

1) An additional membrane marker as a counterstain for JellyOp in Supp. Fig. 1 would be helpful.

2) Why isn't this receptor constitutively active?

Reviewer #3 (Remarks to the Author):

The authors report a new method for activating Gs signaling in a temporal and cardiac specific manner using optogenetics. The experiments are well designed, easy to follow, and experimental

results are clearly establishing that the method works. This new method should be valuable for cardiac researchers. I only have minor comments:

1. Please show the time-course of L-type Ca current activation by light and compare it to acute ISO application. The reported 20% increase seems very small compared to acute b-adrenergic stimulation, which usually increases L-currents by 50% or more.
2. I would encourage the authors to include more figures into the main MS from the supplement to enhance readability of the report.

Point-by-point responses to reviewer comments:

Reviewer #1:

The manuscript reports application of a Gs-coupled opsin (JellyOp) to allow optogenetic control over cAMP concentration and contractile activity in ES cell derived cardiomyocytes and in the intact heart of a transgenic mouse lines. The approach is innovative and the data compelling. The obvious limitation in this dataset is that the technological breakthrough appears not to have been applied to achieve a great advance in understanding of physiology or as a step towards therapy. Concentrating on the characterisation of the optogenetic tool, I found this comprehensive and convincing. JellyOp appears to function very well in this system and give an unprecedented level of control over cardiac activity.

We thank the reviewer for the positive and constructive comments.

I have only a couple of suggestions.

Firstly, The decay in heart rate over 5 min of light exposure is interesting but could have a number of origins. The most important is to distinguish whether it reflects decay of opsin activity vs cell response. I would therefore ask for a side by side comparison of a time course for application beta agonist and or iso. Do changes in heart rate also decay for those treatment?

To answer this point, we have performed additional analyses of the response kinetics and compared our data with inactivation time courses reported in the literature. Side-by-side comparison of application of light vs. the β -agonist isoprenaline was performed in beating hearts. Peak (80%) frequency was observed after ~15 s of illumination, thus during the 90 s illumination period. In contrast, due to perfusion and diffusion delay, isoprenaline showed peak effect only after ~90 s (Fig 5c,d,f), thus inactivation kinetics unfortunately cannot be analyzed during an equally sized 90 s time window. Therefore we characterized inactivation kinetics during illumination and found a time constant of 33.5 ± 18.1 s leading to a steady state level of 65% of the peak effect (see new text on page 7). This time course and the steady state effect fits to that reported for cAMP levels from isolated ventricular mouse cardiomyocytes during continuous isoprenaline stimulation¹. This effect was analyzed in a mathematical model of β -adrenergic signaling and can be explained by phosphodiesterase activity, cAMP fluxes between compartments, and receptor desensitization.² Although we have no conclusive answer how to dissect inactivation of receptor vs. inactivation of downstream signaling, the JellyOp function in the intact heart is very similar to fast β_1 -agonist application to cardiomyocytes in vitro.

My second suggestion is that the authors explore the JellyOp driven response to longer wavelength light. Longer wavelengths penetrate biological tissue more effectively and it would be interesting to know if JellyOp can be activated by them at reasonable intensities.

As suggested, we have performed additional experiments using monochromatic light to determine the spectral sensitivity of JellyOp. We detected heart rate acceleration in the spectral window of 400 to 600 nm with a peak effect at 500 nm (new Fig. 5h). A wavelength of 550 nm showed a ~50% effect, and because it is still heavily absorbed by hemoglobin and myoglobin it does not provide an advantage. Illumination with 600 nm, which is less absorbed, showed only

inconsistent and small responses. Thus, long wavelength stimulation in the near-infrared window (650 – 900 nm)³ is unfortunately not an option for WT JellyOp and would require mutagenesis e.g. by homology modelling to short and long wavelength cone opsins.

Reviewer #2:

This is an interesting manuscript describing a nice way to potential understand Gs signalling in the heart using an optogenetic approach. The authors have developed a system using a novel opsin that can be used to active Gs signalling in cells that express it when stimulated with light. They demonstrate clearly that it activates Gs signalling in cardiomyocytes and in hearts from ES-derived cells and transgenic animals. The experiments are in general well designed and well performed.

We thank the reviewer for the positive and constructive comments.

Some concerns that could addressed:

1) the authors do not demonstrate or know whether JellyOp activates other G proteins. It would be important to tease this out, if they want to claim that it is specific for Gs signalling. Is this an issue for comparing the effects of isoproterenol with JellyOp activation?

To answer this we would like to point out previously published results showing that JellyOp is specific for G_s-activation in HEK293 cells because its effect on MAPK phosphorylation can only be inhibited by blockage of adenylate cyclase but not of G_i-proteins (pertussis-toxin) or phospholipase C (U73122)⁴.

In addition, we obtained similar effects using either JellyOp stimulation with light or isoprenaline-mediated β -receptor activation in embryonic stem cell-derived cardiomyocytes (light: +79%, isoprenaline: +87%, Fig. 2d). Therefore we conclude that JellyOp selectively activates G_s-proteins, but not G_i-activity, which would result in reduced beating frequencies.

It is true that light had a significantly lower effect on Ca²⁺ currents in adult ventricular cardiomyocytes (light: +21%, isoprenaline: +43%) and on heart rate of sinus node cells (light: +44%, isoprenaline: +58%, Fig. 2e). Thus in these cell types, counteracting G_i-activity cannot be entirely excluded, however pertussis-toxin experiments to block G_i are not successful in these cells due to the required long (24 h) incubation time. Another possibility would be privileged signaling of native β -receptors to the G_s-proteins in the structurally more mature cell types compared to an overexpressed receptor from the lower invertebrate prebilaterian animal Jellyfish. We have added these considerations to the revised version of the manuscript (see page 9).

2) Why aren't the kinetics of relaxation affected in muscle cells? This is surprising given that SERCA2 is a target of PKA.

Because of the reviewer's comment, we have performed extensive analysis of relaxation kinetics of isolated ventricular cardiomyocytes. In line with phosphorylation of PLB by PKA disinhibiting SERCA2a function, we found significantly enhanced relaxation kinetics after JellyOp stimulation

with light (new Fig 4 i,k). Interestingly, the delay between onset of illumination (20 s) to the maximal effect on relaxation was ~50% shorter than the maximal effect on fractional cell shortening (new Fig 4l). This very interesting finding on differential signaling kinetics could be explained by privileged phosphorylation of the phospholamban/SERCA2a microdomain compared to the bulk cytosolic phosphorylation as reported earlier.⁵

3) A better transgenic model might have been one that is cardiomyocyte specific. Given that there is limited data associated with arrhythmias in Figure 2 perhaps the connection to disease should be tempered?

This is correct, however in our experience, using cardiomyocyte specific promoters to generate transgenic embryonic stem cells usually adds 1-2 years to a project due to the required extensive differentiation and characterization of cell clones, which is why we have chosen to prove JellyOp function in the heart by ubiquitous overexpression. We would like to stress that this allowed us to provide a proof of principle study on the utility and effectiveness of this technology. In the future, we are planning to use the Cre/LoxP system to generate JellyOp reporter mice for cell-type specific optogenetic G_s-stimulation in cardiomyocytes, fibroblasts, endothelial- and smooth muscle cells, which will allow analysis of paracrine effects of β -adrenergic signaling in the different cell types of the heart.

As suggested, we have added these considerations and tempered the mechanistic conclusions and the potential impact on cardiac arrhythmias in the revised version of the manuscript (see page 8).

Minor issues

1) An additional membrane marker as a counterstain for JellyOp in Supp. Fig. 1 would be helpful.

We have tried to counterstain with the embryonic stem cell membrane marker SSEA1, but unfortunately the specific antibody and the 1D4 rhodopsin antibody are monoclonal mouse IgG1 resulting in cross reaction. To provide better pictures of JellyOp in the respective figure, we have taken high resolution images at Nyquist resolution and performed 3D deconvolution using the Richardson-Lucy Total Variation algorithm⁶ showing clearly membrane-bound JellyOp signals without intracellular staining (new Fig 1b).

2) Why isn't this receptor constitutively active?

JellyOp is not activated by external light in mice, because of the high absorption of light by the chest wall in the sensitive spectral window (450 - 550 nm, new Fig. 5h), This is also confirmed by the lack of cardiac hypertrophy of JellyOp mice (Fig. 4c), which is known to result from constitutive β -adrenergic signaling. This is a bit in contrast to an earlier report claiming little dark activity of JellyOp in HEK293 cells, which could be also due to activation of the very sensitive JellyOp in these experiments by the co-expressed luciferase reporter.⁴

Reviewer #3:

The authors report a new method for activating G_s signaling in a temporal and cardiac specific manner using optogenetics. The experiments are well designed, easy to follow, and experimental results are clearly establishing that the method works. This new method should be valuable for cardiac researchers.

We thank the reviewer for the positive and constructive comments.

I only have minor comments:

1. Please show the time-course of L-type Ca current activation by light and compare it to acute ISO application. The reported 20% increase seems very small compared to acute b-adrenergic stimulation, which usually increases L-currents by 50% or more.

We fully agree with the reviewer and have therefore performed additional patch clamp experiments in JellyOp expressing ventricular cardiomyocytes using isoprenaline to activate β -receptors. In line with the reviewers' comment we found a 43% increase of L-type Ca²⁺ currents, which was significantly higher compared to application of light (21%). This could be either due to counteracting G_i-activity or to the less effective G_s-coupling of overexpressed JellyOp from the lower invertebrate prebilaterian animal Jellyfish compared to the endogenous mammalian β -receptors, especially in mature ventricular cardiomyocytes. We now discuss this result in the revised version of the manuscript (see page 9).

2. I would encourage the authors to include more figures into the main MS from the supplement to enhance readability of the report.

We have followed this good suggestion and incorporated all supplementary figures into the revised main manuscript.

References

1. O'Connell TD, Rodrigo MC, Simpson PC. Isolation and culture of adult mouse cardiac myocytes. *Methods Mol Biol* **357**, 271-296 (2007).
2. Bondarenko VE. A compartmentalized mathematical model of the beta1-adrenergic signaling system in mouse ventricular myocytes. *PLoS One* **9**, e89113 (2014).
3. Golovynskyi S, *et al.* Optical windows for head tissues in near-infrared and short-wave infrared regions: Approaching transcranial light applications. *J Biophotonics* **11**, e201800141 (2018).
4. Bailes HJ, Zhuang LY, Lucas RJ. Reproducible and sustained regulation of Galphas signalling using a metazoan opsin as an optogenetic tool. *PLoS One* **7**, e30774 (2012).
5. Sprenger JU, *et al.* In vivo model with targeted cAMP biosensor reveals changes in receptor-microdomain communication in cardiac disease. *Nat Commun* **6**, 6965 (2015).
6. Sage D, *et al.* DeconvolutionLab2: An open-source software for deconvolution microscopy. *Methods* **115**, 28-41 (2017).

REVIEWERS' COMMENTS:

Reviewer #1 (Remarks to the Author):

I am satisfied with the authors' attempts to address my questions.

I have only one further suggestions. The spectral sensitivity that they now report is consistent to that previously ascribed to JellyOp ($\lambda_{max}=500\text{nm}$), they can then use the opsin nomogram to make accurate predictions of the amount of light required at various wavelengths to achieve the response produced by $10\mu\text{W}/\text{mm}^2$ at 500nm . According to my calculations this would be $400\mu\text{W}/\text{mm}^2$ at 600nm and $2\text{mW}/\text{mm}^2$ at 650nm . They may wish to add this to the description of their result as an indication of the potential for activating the receptor with longer wavelengths.

Reviewer #2 (Remarks to the Author):

Thanks for the thoughtful responses to my queries.

Reviewer #3 (Remarks to the Author):

The authors have addressed all of my concerns. Excellent work!

Point-by-point responses to reviewer comments:

Reviewer #1

I am satisfied with the authors' attempts to address my questions.

I have only one further suggestions. The spectral sensitivity that they now report is consistent to that previously ascribed to JellyOp ($\lambda_{max}=500\text{nm}$), they can then use the opsin nomogram to make accurate predictions of the amount of light required at various wavelengths to achieve the response produced by $10\mu\text{W}/\text{mm}^2$ at 500nm . According to my calculations this would be $400\mu\text{W}/\text{mm}^2$ at 600nm and $2\text{mW}/\text{mm}^2$ at 650nm . They may wish to add this to the description of their result as an indication of the potential for activating the receptor with longer wavelengths.

We thank the reviewer for this constructive comment. As suggested we have fitted the values with the Govardovskii nomogram¹ which results in a peak wavelength of 493 nm , which is in accordance with the reported action spectrum of wild-type JellyOp². We also comment toward using longer wavelength and add the proposed calculations. However we also would like to mention that these calculations are not based on experimental results and might be misleading because especially light with longer wavelengths will penetrate more efficiently into myocardial tissue. On the other site, increasing light intensities will be limited by toxic effects.

Reviewer #2

Thanks for the thoughtful responses to my queries.

Reviewer #3

The authors have addressed all of my concerns. Excellent work!

References

1. Govardovskii VI, Fyhrquist N, Reuter T, Kuzmin DG, Donner K. In search of the visual pigment template. *Vis Neurosci* **17**, 509-528 (2000).
2. Gerrard E, *et al.* Convergent evolution of tertiary structure in rhodopsin visual proteins from vertebrates and box jellyfish. *Proc Natl Acad Sci U S A* **115**, 6201-6206 (2018).